# DenseFormer: Enhancing Information Flow in Transformers via Depth Weighted Averaging

**Matteo Pagliardini**[*]
EPFL
matteo.pagliardini@epfl.ch

**Amirkeivan Mohtashami**[*]
EPFL
amirkeivan.mohtashami@epfl.ch

**Francois Fleuret**
University of Geneva
Francois.Fleuret@unige.ch

**Martin Jaggi**
EPFL
martin.jaggi@epfl.ch

## Abstract

The transformer architecture by Vaswani et al. [31] is now ubiquitous across application domains, from natural language processing to speech processing and image understanding. We propose DenseFormer, a simple modification to the standard architecture that improves the perplexity of the model without increasing its size—adding at most a few thousand parameters for large-scale models. Our approach relies on an additional averaging step after each transformer block, which computes a weighted average of current and past representations—we refer to this operation as Depth-Weighted-Average (DWA). The learned DWA weights exhibit coherent patterns of information flow, revealing the strong and structured reuse of activations from distant layers. Experiments demonstrate that DenseFormer is more data efficient, reaching the same perplexity of much deeper transformer models, and that for the same perplexity, these new models outperform transformer baselines in terms of memory efficiency and inference time.

## 1 Introduction

The transformer architecture [31] is the workhorse of modern natural language processing. Recent leaps in the state of the art can be attributed in a large part to efforts scaling this architecture, from millions of parameters [3] to large billion-parameter models [1, 18, 23, 29, 30]. Unfortunately, those larger models come with an increased computational cost, and a large memory footprint. This renders them impractical to use in a wide range of use-cases, limiting who can benefit from them to a handful of big corporations. As an attempt to mitigate this issue, Touvron et al. [29] propose training a smaller model for more steps. However, longer training requires larger datasets which becomes challenging as we are reaching scales where even extremely large datasets fall short of sufficient amounts of data [32].

Furthermore, recent observations suggest that we are reaching the state of diminishing returns where increasing the depth of the model beyond a certain point does not significantly improve performance [21]. Interestingly, a similar state of diminishing returns has been observed in the field of computer vision focused on the training of deep convolutional neural networks. Various solutions were proposed to address this issue, including DenseNets [11] which alleviated the problem by allowing subsequent layers to directly access outputs of earlier layers.

In this work, using a similar intuition as DenseNets, we propose the DenseFormer architecture. In particular, instead of only having skip connections from one block to the next, in DenseFormer, a

---

[*]Equal contribution.

38th Conference on Neural Information Processing Systems (NeurIPS 2024).

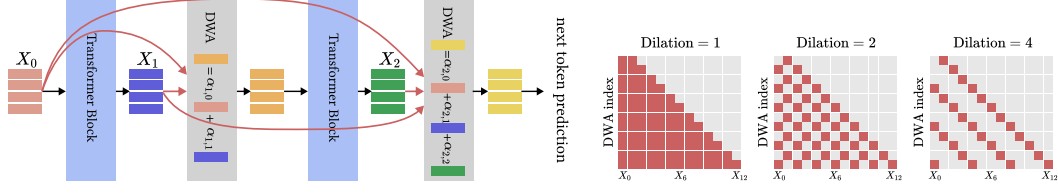

(a) DenseFormer Architecture.

(b) DWA Weights with Dilation.

Figure 1: **DenseFormer architecture.** The diagram in **(a)** shows the DenseFormer architecture with two transformer layers and a dilation of 1. After the first (resp. second) block, the past and current intermediary representations $\{X_0, X_1\}$ (resp. $\{X_0, X_1, X_2\}$) are averaged using the first (resp. second) DWA weights $[\alpha_{1,0}, \alpha_{1,1}]$ (resp. $[\alpha_{2,0}, \alpha_{2,1}, \alpha_{2,2}]$). The DWA weights are supported by red arrows. Those weights are represented in matrix form in **(b)**, for a 12 layers DenseFormer. A DWA module at depth $i$ has $i + 1$ weights, represented in red. Increasing the dilation sparsifies this matrix, reducing the computational overhead without degrading the perplexity, see Section 3.2 for more details.

weighted average of the outputs of all previous blocks is given as the input of the next block. The approach is visually summarized in Fig. 1a.

We show that DenseFormers can perform the same as a much deeper standard Transformer model while at the same time being smaller in size, faster, and consuming less memory at inference. More importantly, this is achieved without increasing the amount of required data. As such, DenseFormers are also more data efficient, obtaining much better performance when trained on the same amount of data than a standard model with a similar number of parameters. Our results establish the DenseFormer architecture as an improved version of Transformers for language modeling, encouraging their future use.

In addition to providing experimental results on DenseFormer's performance, we also provide additional insights and intuition for their success. Looking at the learned weights of the DWA modules we observe a surprisingly stable pattern in the learned weights that emerges at multiple depths and generalizes across random seeds (see Fig. 4). Overall, similar to Huang et al. [11], we hypothesize that the inter-block connectivity enables the model to more directly re-use early features, without requiring to allocate as much bandwidth to propagate them through multiple layers. Intuitively, this seems to help resolve an ambiguity caused by skip-connections which force the deeper representations to maintain the current token representation while at the same time having to predict the next token.

**Contributions.** Our contributions can be summarized as follows:

- Introducing DenseFormer architecture by adding a depth-weighted-average module after each Transformer block.

- Demonstrating over different settings (e.g. datasets, batch sizes, sequence lengths) the significantly superior performance of DenseFormer over deeper Transformers, yielding a better speed-performance trade-off during both inference and training.

- Providing additional empirically grounded insights and intuition that support the benefits of using DenseFormer.

## 2 Related Work

While larger models have shown great promise in delivering better capabilities, recent results suggest that the gain from using deeper models faces diminishing returns [22]. Interestingly, the same challenge presented itself earlier for scaling convolutional neural networks [9]. Methods that allow a better flow of information from earlier to later layers such as Residual connections [10] and Highway Networks [27] have been proposed as a successful solution to tackle this challenge, stabilizing training and increasing the threshold where a gain can be observed from increasing depth. Taking it to the extreme, DenseNets [11] demonstrate the benefit of having access to the output of all previous layers. We propose DenseFormer by building on a similar intuition, allowing each block to directly access the output of all previous blocks.

Some advantages of attending to representations from earlier layers have already been explored in prior work. Depth-wise Attention [4] suggests adding an attention-like layer before Transformer's

final projection layer. This new layer applies attention across the outputs of the Transformer's blocks for the current token (instead of over different tokens as in the standard attention layer). This operation is similar to the weighted averaging block introduced in this work which mixes the outputs of earlier blocks. However, whereas in our proposal the weights are learned during training, Depth-wise Attention computes the weights using the dot product (similar to the attention) which levies a higher overhead. In our experiments, we also show that only a single DWA step before the last layer does not yield the same performance as a full DenseFormer. In another recent relevant work Mohtashami et al. [17] suggest interleaving current and past representations. This allows current tokens to attend to previous representations of themselves and past tokens. Our DenseFormer can be seen as a crude and much more efficient approximation of this mechanism in which we restrict each token to only attend to past representations of themselves, using static (as opposed to dynamic) attention weights. Most relevant to us—in the context of low data training—the concurrent work of Charpentier & Samuel [2] also adds links from the input of each block to representations from earlier layers. An important difference is their use of softmax weights. We conjecture it is important to let the model subtract representations using negative weights (see § 5), and implementing their methods only outperformed the baseline in a low-data regime (see App. B.7). Moreover, they do not provide an efficient implementation of their method.

Since its original design by Vaswani et al. [31], the Transformer architecture used in most applications changed surprisingly little. LLM training is costly, and architecture choices are often conservative. Small variations include changing the activation function [26], adopting RMSNorm instead of LayerNorm, or computing the feed-forward and attention in parallel [20, 29, 33]. More progressive proposals have been made to alleviate computational challenges of the self attention module, such as using kernel methods or other linear approximations [12, 13, 34], or removing redundant operations without impacting performance [8, 25]. These proposals only affect the internal structure of Transformer blocks. As DenseFormer only adds DWA modules that operate between blocks, we expect that it can be readily used together with these existing proposals.

Recent explorations also have shown gains from using multiple language models instead of one. An example is using a mixture of experts, which rely on a routing mechanism [5] to select which expert(s) to use in a given context. Other examples include deploying the same instance of a model in different roles allowing them to debate or provide feedback to each other leading to performance improvements [14, 16]. As these approaches mostly retain the structure of the Transformer architecture and focus on the communication structure between multiple models (or sub-modules), we also expect them to be adaptable to use DenseFormers.

## 3 Method

**Setup & Notations.** We consider the standard Transformer architecture. Given a depth $d$, it consists in a succession of $d$ Transformer blocks $B_1, \ldots, B_d$, each composed of a self-attention module followed by a single hidden layer Multi-Layer-Perceptron (MLP). We name $X_0, \ldots, X_d$ the different intermediary representations, with $X_0$ being the embedded token sequence, and $X_i$ for $i \geq 1$ being the output of block $B_i$.

We summarize the Transformer architecture as follows:

$$X_0 := \text{Embedding}(X)$$
$$\forall i = 1, \ldots d, \ X_i := B_i(X_{i-1})$$
$$\text{Transformer}(X) := X_d.$$

**DenseFormer.** The only change to the original architecture is the addition of a **Depth Weighted Average module (DWA)** after each transformer block. A DWA module at depth $i$ performs a weighted average between (i) the output from the current block $B_i$, (ii) the output of all previous blocks $B_{j<i}$, and (iii) the embedded input $X_0$. The weights of the weighted-average for the $\text{DWA}_i$ module at depth $i$ are $\alpha_{i,0}, \ldots, \alpha_{i,i}$. A visual summary can be seen in Fig 1a. The elements of the $\alpha$ matrix are the only additional parameters of our method. More formally, our DenseFormer model can be

summarized as follows:

$$X_0 := \text{Embedding}(X)$$
$$Y_0 := X_0$$
$$\forall i = 1, \ldots d, \ X_i := B_i(Y_{i-1})$$
$$\forall i = 1, \ldots d, \ Y_i := \text{DWA}_i(\{X_0, \ldots, X_i\}) = \sum_{j=0}^{i} \alpha_{i,j} \cdot X_j$$
$$\text{DenseFormer}(X) := Y_d.$$

In Section 4 we demonstrate that the DenseFormer architecture can outperform the standard Transformer architecture. In particular, it obtains a much stronger performance (in terms of perplexity) than a model of the same depth, matching the performance of a much deeper model which is both slower at inference and larger in size, leading to a much higher memory footprint than DenseFormer. We further demonstrate the importance of the improved inter-block connectivity brought by the DWA modules in Section 5. We do so by comparing our architecture to a variety of baselines with constrained connections and show those do not perform as well as DenseFormers.

**Initializing the DWA modules.** We note that if $\alpha_{i,i}$ is set to $1$ while others are set to $0$, the DWA module acts as an identity function, reducing DenseFormer to the standard Transformer architecture. Therefore, we start our training from this initialization.

### 3.1 Impact on Resources

**Negligible Model Size Overhead.** At depth $i$ the DWA module has $i + 1$ weights. Therefore, for a DenseFormer of depth $d$, the total number of additional parameters is $\sum_{j=1}^{d}(j+1) = \frac{d(d+3)}{2}$. For typical model depths (less than 100 blocks), this represents at most an order of $10^3$ parameters, which is negligible when compared to the full size of the models.

**Negligible Memory Overhead.** We also emphasize that while DWA requires access to the output of blocks and embedded input $X_0, \ldots, X_d$, these values are stored even when using the standard Transformer architecture. During training, the outputs of the blocks are kept in memory to allow backpropagation, while at inference, only the blocks' outputs for the current token need to be stored. Therefore, the total memory overhead of DenseFormer is negligible.

**Computational Overhead.** Computing the output of the DWA modules increases the computational cost since it requires averaging over multiple large tensors of size (batch size $\times$ sequence length $\times$ hidden dimension). In this work, we provide an efficient implementation of DWA to reduce the overhead and avoid unnecessary data movement. In addition, we introduce two architectural hyperparameters, which allow building a set of DenseFormer variants that approximate the full DenseFormer. These hyperparameters are DWA dilation and DWA periodicity, respectively introduced in Sections 3.2 and 3.3. We refer to a DenseFormer variant with the dilation factor $k$ and the DWA periodicity $p$ as $k$x$p$-DenseFormer. In this notation, the full DenseFormer is a $1x1$-DenseFormer.

### 3.2 Dilated DenseFormer

In order to further reduce the computational overhead, we introduce a dilation parameter which sparsifies the DWA weights by periodically setting them to 0. In particular, each DWA module is now given the output of every $k$-th block, where $k$ is called the dilation factor. More formally, given a DWA module at depth $i$, a dilation factor of $k$ implies $\text{DWA}_i$ is only computing a weighted average over $\{X_j | j \leq i, j \equiv i \pmod{k}\}$). See Fig. 1b for a visual explanation. Our dilated DenseFormer can be described as:

$$X_0 := \text{Embedding}(X)$$
$$Y_0 := X_0$$
$$\forall i = 1, \ldots d, \ X_i := B_i(Y_{i-1})$$
$$\forall i = 1, \ldots d, \ Y_i := \text{DWA}_i^{\text{dilated}}(\{X_j | j \leq i, j \equiv i \pmod{k}\})$$
$$\text{DenseFormer}(X) := Y_d.$$

As shown in Section 4, we observe no noticeable performance degradation for small values of $k$ (e.g. 2 or 4) while the computational overhead is significantly reduced, leading to much faster inference

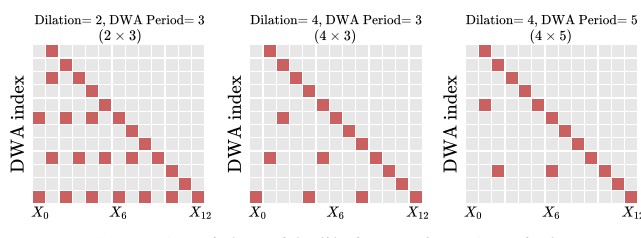

(a) DWA weights with dilation *and* DWA period.

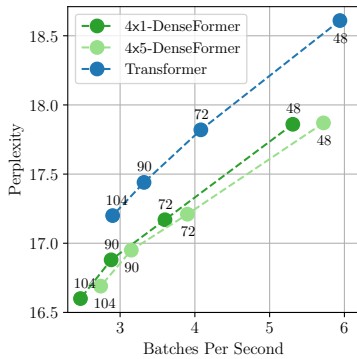

(b) Speed and performance trade-off.

Figure 2: **(a): DWA weights with dilation *and* DWA period:** For a 12 layers DenseFormer, the $\alpha$ weights are sparsified using dilation $k$ and DWA periodicity $p$ (referred to as $k\mathsf{x}p$). Compared to Fig. 1b, only certain rows have some weights other than the upper diagonal weights (which correspond to the regular transformer information flow). Increasing the dilation and period sparsifies the $\alpha$ matrix, reducing the computational overhead without degrading the perplexity, see Sections 3.2 and 3.3 for more details. **(b): Speed and performance trade-off:** Comparison of speed and performance trade-off between the standard Transformer architecture and DenseFormer. The number of blocks in each architecture is reported next to the data-point. All DenseFormer models on this plot use a dilation factor of 4. We show results using a DWA period of 1 and 5. **Comparing perplexities:** Considering only the perplexity (y-axis), a 48 block DenseFormer performs similarly as a much deeper 72 block Transformer. **Comparing trade-offs:** A 48 block 4x5-DenseFormer matches the better perplexity of a 72 block Transformer while being $1.4\times$ faster at inference.

and training. More precisely, a dilation of $k$ reduces the computational overhead induced by DWA modules by a factor of $1/k$.

### 3.3 Periodic DenseFormer

An alternative method to dilation for sparsifying DWA weights is adding DWA modules to the architecture less frequently. In particular, we can consider only adding the DWA after every $p$ blocks (instead of after every block as in the standard DenseFormer). We refer to $p$ as the DWA period. A standard DenseFormer has period 1. A DenseFormer with dilation $k$ and DWA period $p$—referred to as $k\mathsf{x}p$-DenseFormer—can be formalized similarly as a DenseFormer with the following distinction:

$$\forall i = 1, \ldots d, \; Y_i := \begin{cases} \mathrm{DWA}_i^{\mathrm{dilated}}(\{X_j | j \leq i, j \equiv i (\mathrm{mod}\ k)\}) & p \mid i \\ X_i & p \nmid i \end{cases}$$

A visual representation of the matrix of $\alpha$ weights can be seen in Fig. 2a. By increasing the periodicity $p$, we further reduce the computational cost of DenseFormer. In Section 4 we evaluate the effect of increasing the period for a 4-dilated DenseFormer on both performance and speed. We observe that using small values larger than 1 for the period can provide a noticeable boost in speed without noticeable performance degradation. Using a period of $p$ reduces the computational overhead by a factor of $1/p$. Hence, a $k\mathsf{x}p$-DenseFormer only has $1/kp$ of the computational overhead of a regular DenseFormer. In Section 5 we also provide results with other sparsity patterns for the DWA weights ($\alpha$) but show that using dilation and periodicity works most favorably.

**Interplay between $k$ and $p$.** The ideal value of $p$ can depend on the dilation $k$ used. For instance, using $k = p = 4$ implies the DWA module after block 4 will look at $\{X_0, X_4\}$. Its output, $Y_4$, will be sent through block 5 to yield $X_5$. However, the next DWA module after block 8 will only look at $\{X_0, X_4, X_8\}$ (and not at $X_5$). This means that $Y_4$ will have to go through blocks $5, 6, 7, 8$ before being accessible by later DWA modules. In contrast, using $k = 4$ and $p = 5$ allows the information to propagate much faster since DWA modules always have access to the processed output of the previous DWA module. This interplay can be visualized in Fig. 2a as well as in Appendix B.2.

Table 1: **Performance of DenseFormer and the standard architecture of different sizes on OpenWebText2 dataset.** The number of millions of parameters is reported as well as the final perplexity. Additionally the number of batches of size 64 that can be processed in one second is reported as a measure of inference speed. The results are based on three runs with different seeds. The mean value is reported with the standard error reported in parenthesis. In terms of perplexity, DenseFormer clearly outperforms a standard Transformer of the same depth as well as standard Transformers with a similar inference speed. While sometimes a deeper model with the standard architecture can match the performance of a shallower DenseFormer (e.g. 72 block standard architecture and 48 block DenseFormer), inference using the shallow DenseFormer remains much faster. The inference speed is significantly improved with negligible effect on perplexity when increasing the dilation factor and DWA period. Adding a scaling factor to all skip connections in the standard architecture (named Skips with Gains) does not yield the same performance boost as DenseFormer highlighting the importance of inter-block connectivity in DenseFormer.

| Model | Dilation×Period | Depth | Parameters # (M) | Perplexity ($\downarrow$) | Inference BPS ($\uparrow$) |
|---|---|---|---|---|---|
| Transformer | - | 48 | 378.45 | 18.61 (0.02) | 5.94 (0.00) |
| Skips With Gains | - | 48 | 378.45 | 18.45 (0.03) | 5.72 (0.01) |
| | $1 \times 1$ | 48 | 378.45 | 17.84 (0.00) | 4.65 (0.00) |
| DenseFormer | $4 \times 1$ | 48 | 378.45 | 17.86 (0.02) | 5.31 (0.01) |
| | $4 \times 5$ | 48 | 378.45 | 17.87 (0.02) | 5.72 (0.00) |
| Transformer | - | 64 | 491.72 | 17.94 (0.01) | 4.57 (0.00) |
| | - | 72 | 548.35 | 17.82 (0.04) | 4.08 (0.00) |
| | $1 \times 1$ | 72 | 548.36 | 17.12 (0.02) | 2.93 (0.00) |
| DenseFormer | $4 \times 1$ | 72 | 548.35 | 17.17 (0.00) | 3.60 (0.00) |
| | $4 \times 5$ | 72 | 548.35 | 17.21 (0.01) | 3.90 (0.00) |
| Transformer | - | 84 | 633.31 | 17.48 (0.01) | 3.54 (0.00) |
| | - | 90 | 675.78 | 17.44 (0.01) | 3.32 (0.00) |

## 4   Results

We demonstrate the effectiveness of DenseFormer through experiments on language modeling tasks. We compare the performance of DenseFormer architectures with the standard Transformer architecture using model size, inference time, training time, and final perplexity (sometimes abbreviated as PPL) as metrics. For each metric, we consider a baseline that performs the same as DenseFormer on that metric. Concretely, we include the following baselines:

**Same Depth Baseline.** A standard architecture with the same depth as the DenseFormer. It roughly has the same number of parameters as DenseFormer given the negligible number of DWA parameters.

**Same Inference Time Baseline.** A standard architecture that has the same inference time as the DenseFormer. Since adding DWAs to the architecture has a computational overhead, this baseline has more layers (i.e. more capacity) than DenseFormer.

**Same Perplexity Baseline.** A standard architecture with roughly the same PPL as the DenseFormer. It usually is much deeper than the DenseFormer, showcasing the benefits of using DWAs.

**Same Training Time Baseline.** A standard architecture that has the same training time as the DenseFormer. Since adding DWAs to the architecture has a computational overhead, this baseline is trained for more iterations than DenseFormer.

**Skips with Gains.** It can be observed that DenseFormer, among other things, allows scaling the output of the previous layer, providing more control than the original skip connections. Therefore, we provide an additional baseline to show this is not the only benefit offered by DenseFormer and emphasize the importance of having direct access to the outputs of earlier layers. In particular, we consider a modified version of the standard architecture where each skip connection also contains a learned scaling factor which is applied to the the values moving through the skip connection before being summed with the output from a different layer (e.g. self-attention).

**Experimental setup.** We use models with 8 heads, each having 64 dimensions, and train them using batches of 400 sequences of length 256. We use rotary positional encoding [28]. We optimize the model using AdamW [15] with $\beta_1 = 0.9$ and $\beta_2 = 0.95$ with weight decay factor 0.1.

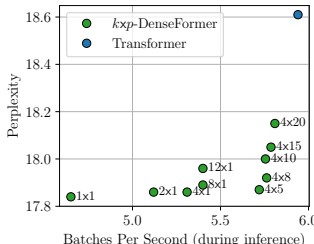 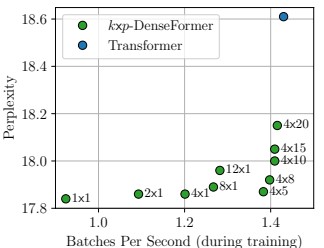 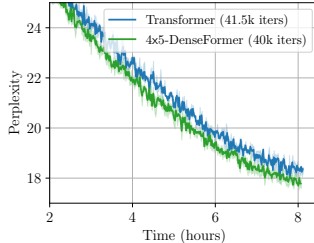

(a) PPL vs. Inference speed (BPS)  (b) PPL vs. Training speed (BPS)  (c) Comparison under same training time

Figure 3: **Training and inference efficiency of $k$x$p$-DenseFormer vs. Transformer.** For 48 block models, we compare in **(a)** the different perplexity/inference speed trade-offs reached by a regular Transformer and $k$x$p$-DenseFormers. In the top right corner, the Transformer baseline is the model with the worst perplexity but the fastest at inference. In contrast, the 1x1-DenseFormer in the bottom left corner, is reaching the best perplexity but incurs a cost in inference speed. By varying the dilation $k$ and DWA period $p$, some $k$x$p$-DenseFormer models (e.g. 4x5) provide most of the perplexity improvement of the original DenseFormer while significantly reducing the time overhead. A similar analysis holds when looking at the training speed in **(b)**. In **(c)**, we show the perplexity decreasing during training. The x-axis is time. To compensate for the computational overhead of DenseFormer, we train the Transformer baseline for more iterations, such that the two methods have the same training time budget. We observe how our 4x5-DenseFormer is reaching a better perplexity faster than the baseline.

We perform most of our experiments on the OpenWebText2 dataset [6], an enhanced version of OpenWebTextCorpus [7] with around 17B tokens. We train all models for $40k$ steps, thus keeping the number of data points used in training fixed. We use learning rate 0.001 with a cosine scheduler and do a warmup in the beginning 5% steps.

We present the result of training 48 block and 72 block DenseFormers along with baselines of various sizes in Tab. 1. We make the following observations based on these results:

**Better perplexity than same depth baseline.** When comparing with a baseline of the same depth, DenseFormer significantly outperforms the standard architecture. Moreover, as it can be seen in Fig. 2b, the perplexity of a 48 block DenseFormer is only matched by a 72 block Transformer baseline.

**Faster than a baseline with the same perplexity.** The performance of a 48 block DenseFormer is on par with a 72 block standard architecture. Still, the 48 block DenseFormer is much faster at inference (measured in batches per second) than the 72 block standard architecture. Moreover, the number of parameters and memory footprint of the 72 block baseline is 45% larger than the one of the 48 block DenseFormer.

**Better perplexity than a baseline with the same inference time.** Comparing the 48 block Dense-Former without dilation, with a 64 block standard architecture (which has the same inference speed), shows a wide gap between the higher performance of DenseFormer (17.84) and the standard Transformer architecture (17.94). Considering DenseFormer models with a dilation of 4 and/or a DWA period of 5 would increase this gap further.

**Weighted skip-connections are insufficient.** DenseFormer is changing the flow of information in the model. We can wonder whether it leverages the additional expressivity or whether the performance gains could be explained by making it easier to rescale the contribution of each block. When comparing the 48 block Transformer baseline to the 48 block skip-with-gains baseline, it seems adding tunable weights to each skip connection does not lead to a significant improvement. When compared with the 48 block DenseFormer, this showcases the importance of having direct access to all previous layers.

**Faster with dilation and DWA period.** Finally, our Tab.1 results show that for small dilation factors $k$ and DWA period $p$, $k$x$p$-DenseFormer perform comparably while significantly boosting speed. Indeed, as can also be seen in Fig. 2b, using 4x1-DenseFormers or 4x5-DenseFormers allows pushing the Pareto frontier on speed and performance trade-off forward.

**More efficient during training.** We train a 48 block 4x5-DenseFormer and compare it against a 48 block Transformer baseline trained *with the same time budget*. The baseline is therefore trained for more iterations (41.5k vs. 40k) to compensate for the time overhead of DenseFormer. In Fig. 3c we

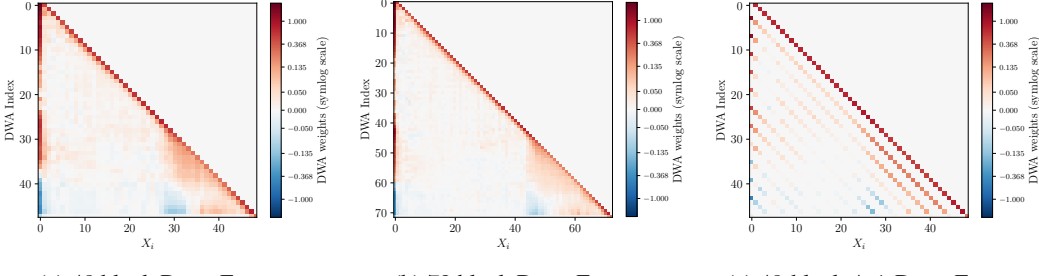

| (a) 48 block DenseFormer. | (b) 72 block DenseFormer. | (c) 48-block 4x1-DenseFormer. |

**Figure 4: Visualization of DWA Learned Weights.** Each row shows the weights $\alpha$ learned by a DWA module at a given depth. While the heatmaps are averaged across 3 runs with different seeds, those patterns are very consistent across seeds. In **(a) and (b)**, strikingly similar patterns can be observed in both 48 and 72 layer DenseFormers. In **(c)**, we show the learned weights for a 48 block DenseFormer trained with a dilation of 4. Despite the sparsity, we still observe a very similar pattern to those learned by the non-dilated models.

Table 2: **Same training time comparison.** Comparison of 4x5-DenseFormer's performance against a standard Transformer trained for more iterations. The number of training steps of the standard architecture is chosen such that the training time is roughly the same (and always more than) that of the DenseFormer. Both architectures have 48 blocks and are trained with 2000 warmup steps. Even though the Transformer is trained with more steps, it is still outperformed by the DenseFormer.

| Model | Steps | Train time (h) | Perplexity |
|---|---|---|---|
| Standard | 41500 | 8.09 | 18.33 (0.00) |
| 4x5-DenseFormer | 40000 | 8.04 | **17.87 (0.02)** |

visualize the perplexity (approximated on a small subset of the validation set) dropping as a function of the training time. The DenseFormer's perplexity is dropping faster than that of the Transformer baseline. This shows the superior efficiency of DenseFormer during training. While the Transformer is trained for more steps, thus using more data points, it is still outperformed by the DenseFormer. The final perplexities on full validation set reached by the two models can be seen in Tab. 2.

## 4.1 Additional Experiments

We perform additional experiments to show that our results are general and extend to different settings and larger scales. We also study the impact of the dilation factor $k$ and DWA period $p$ on the efficiency of our $k$x$p$-DenseFormer architecture.

**Experiments with longer sequences.** Due to computation limitations, we can not run all experiments at a large scale. We however repeat a limited set of experiments with longer sequences of 512 tokens using a smaller batch size of 128. A 48 block Transformer baseline reaches a final perplexity of $18.28 \pm 0.03$ against $17.73 \pm 0.02$ for the DenseFormer. This result shows that the gap between the two architectures persists for longer sequences.

**Effect of Dilation and DWA period.** Fig. 3a and Fig. 3b show the impact of using different combinations of dilation $k$ and DWA period $p$ on the final perplexity, training and inference speed. As can be seen, small values of the dilation factor (e.g. up to 4) have a negligible effect on the perplexity. However, increasing the dilation factor further affects the performance more adversely while the gain in both training and inference speed starts to plateau. Increasing the DWA period also provides a similar trade-off, with the perplexity being barely affected for $p \leq 5$. From those figures, we conclude that a dilation of 4 and a DWA period of 5 seem to offer the best compromise between speed and perplexity. In Appendix B.2, we provide more detailed results, including showing how increasing dilation yields a more pronounced speed-up for deeper models, making larger dilation factors more effective in those scales.

**PG-19 experiments.** We also show the superior performance of DenseFormer on the PG-19 dataset [24]. At any depth, DenseFormers outperform Transformers in PPL. See App. B.3 for details.

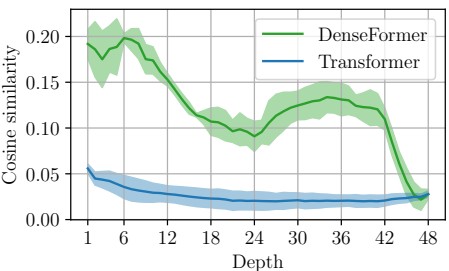

Figure 5: **Cosine similarity between the output of each DWA module and the initial embedding vectors.** The results are averaged over three seeds, for DenseFormer models with 48 blocks and no dilation (corresponding to the weights in Fig. 1a). The model initially maintains a high correlation between the input and the output of each DWA module. That correlation decreases in later layers. Intuitively, we can hypothesize that this is the stage where the model is preparing to output the next token. A very similar plot can be observed for 72 block models in Appendix B.1.

Table 3: **Alternative DWA Sparsity Patterns.** We compare 48 block architectures with different sparsity patterns. In Last K, DWA can only access the output of the last $k$ blocks as well as the input embedding vectors. Connect to Last includes only a single DWA module in the architecture placed after the last layer, i.e. connecting every block to the last block. Neither of these patterns achieves the same perplexity boost as our proposed Dense-Former. 12x1 and 4x5-DenseFormers—with sparsities of respectively 92% and 95%—outperform other sparsity patterns, which signifies the importance of pairwise inter-block connections.

| Sparsity Pattern | Perplexity | Sparsity |
|---|---|---|
| Baseline Transformer | 18.61 (0.02) | - |
| Last K ($K = 4$) | 18.28 (0.01) | 84% |
| Connect to Last | 18.33 (0.02) | 96% |
| 12x1-DenseFormer | 17.96 (0.01) | 92% |
| 4x5-DenseFormer | **17.87 (0.02)** | 95% |

## 5 Analyzing the Information Flow

In this section, we investigate the learned DWA $\alpha$ weights to gain more insight and intuition on the reason behind the superiority of DenseFormer.

**A stable weight pattern emerges.** We start by visualizing the learnt DWA weights for 48 and 72-block DenseFormer models (with a dilation and period of 1) in Fig. 4. Interestingly, the $\alpha$ weight patterns learned at both depths are very similar:

- High weights are on the diagonal (corresponding to the normal information flow as in a standard Transformer) as well as on the immediate previous blocks.

- High weights are given to the initial embedding vectors. Those weights are positive in earlier layers while later layers assign a negative weight.

- An aggregation block is observed near the final layers where a high weight is given to all previous layers in the block (seen as a high-weight triangle near the diagonal in the lower right corner).

Finally, Fig. 4c shows similar patterns persist to some extent when using dilation factors higher than 1. Similar results hold for $k$x$p$-DenseFormers as can be seen in Appendix B.1.

**Other sparsity patterns during training.** Alternative to pruning after training, we also consider imposing different sparsity patterns during training. We already have experimented with such sparsity patterns induced through dilation and DWA period. In Fig. 4, we observe that in many cases the largest weights are given to the few previous blocks as well as the first block. As such, we experiment with a sparsity pattern allowing DWA to access previous $k$ blocks as well as the input embedding vectors calling it "Last K". Furthermore, given the large magnitude of weights on the last layer, we also experiment with only having a single DWA in the architecture which is placed after the last layer, calling it "Connect to Last". Tab. 3 shows the perplexities when using these sparsity patterns and shows that they do not achieve the boost in performance obtained with DenseFormer. This observation further strengthens the importance of small DWA weights both during training and inference.

**Correlation with Input Embeddings.** Based on the special pattern of weights given to the embedding vectors—especially the negative weights given to the input by the last layers (see Fig. 4)—we hypothesize that the model tries to use the information in the input in earlier layers while removing the influence of the current token as it tries to predict the next token. In order to test this hypothesis we plot the average cosine similarity of each token's vector after each DWA block with its input embedding in Fig. 5. As expected based on the weight pattern, we observe that the similarity is high in the earlier layers. At the later stage, the model decreases this similarity significantly.

We hypothesize this final decrease is due to the model moving from processing the current token to building the next token's representation. In contrast, the similarity drops down very early for a standard transformer and remains low for the rest of the layers.

## 6   Future Work & Conclusion

In this paper, we introduced the DenseFormer architecture. This architecture adds an averaging module called DWA after each block which allows it to directly access the outputs of previous blocks. We established the superiority of this architecture over Transformers in terms of perplexity/speed trade-off through experiments in a variety of settings. Additionally, we provided dilation and DWA periodicity as simple methods to improve speed without significantly hurting performance. Finally, we provided insights about the learned weights, revealing patterns persisting over different depths.

As the next steps, finding more efficient implementations of DenseFormer is grounds for future work. One possible direction is finding better sparsity patterns that also can be implemented efficiently. The weights visualization in Fig. 4 suggests such patterns might exist. Furthermore, finding efficient ways to shard DWA across multiple nodes is important to allow large-scale distributed training.

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

# Contents

# A Implementation details

In this section, we provide several implementations in Pytorch [19]. The first implementations we propose are very simple and rely on looping over the past representations to compute each DWA output. As a result, those naive implementations are slow when dilation and DWA periodicity are not used. In a second time, we propose a more optimized and faster implementation. As a comparison, for a 4x5-DenseFormer with 48 blocks, the naive implementation takes 674ms per training iteration against 657ms for the later implementation. In Appendix A.2, we provide a denseformer python package to help turn Transformers into DenseFormers in only 3 simple steps.

**TL;DR:** If you want to implement your own DenseFormer, simply check the denseformer python package: https://github.com/epfml/DenseFormer

## A.1 Naive Pytorch implementation

**Naive 1x1-DenseFormer implementation.** A naive implementation in Pytorch would consist of storing the output of each block during the forward, and feeding those representations to DWA modules after each block. A pseudocode would look like the following:

Listing 1: Naive 1x1-DenseFormer implementation

```python
import torch

class DWA(torch.nn.Module):

  def __init__(self, n_alphas):
    super().__init__()
    self.n_alphas = n_alphas
    alphas = torch.zeros((n_alphas,))
    alphas[-1] = 1.0
    self.alphas = torch.nn.Parameter(alphas)

  def forward(self, all_previous_x):
    weighted_avg = all_previous_x[0] * self.alphas[0]
    for i in range(1, self.n_alphas):
        weighted_avg += self.alphas[i] * all_previous_x[i]
    return weighted_avg

class GPTBase(torch.nn.Module):

  def __init__(self, config):
    super().__init__()
    self.config = config
    self.dwa_modules = torch.nn.ModuleList([DWA(n_alphas=i+2) for i in range(config.n_blocks)])
    self.wte = torch.nn.Embedding(config.vocab_size, config.n_embd)
    self.blocks = torch.nn.ModuleList([Block(config) for _ in range(config.n_blocks)])
    self.ln_f = LayerNorm(config.n_embd, bias=config.bias)
    self.lm_head = torch.nn.Linear(config.n_embd, config.vocab_size, bias=False)
    self.transformer.wte.weight = self.lm_head.weight # weight tying

  def forward(self, idx):
    x = self.wte(idx)
    all_previous_x = [x] # This stores all the intermediary representations
    for i in range(self.config.n_blocks):
      x = self.blocks[i](x)
      all_previous_x.append(x)
      x = self.dwa_modules[i](all_previous_x) # Computing the weighted average
    x = self.ln_f(x)
    logits = self.lm_head(x)
    return logits
```

**Naive $k$x$p$-DenseFormer implementation.** Here we introduce a naive implementation with dilation and DWA-frequency. The DWA module in the following implementation would be the same as for the 1x1-DenseFormer naive implementation.

Listing 2: Naive $k$x$p$-DenseFormer implementation

```python
class GPTBase(torch.nn.Module):

  def __init__(self, config):
    super().__init__()
    self.config = config
    self.dwa_modules = torch.nn.ModuleList([DWA(n_alphas=(i+1+config.dilation)//config.dilation)
      for i in range(config.n_blocks)])
    self.wte = torch.nn.Embedding(config.vocab_size, config.n_embd)
    self.blocks = torch.nn.ModuleList([Block(config) for _ in range(config.n_blocks)])
    self.ln_f = LayerNorm(config.n_embd, bias=config.bias)
    self.lm_head = torch.nn.Linear(config.n_embd, config.vocab_size, bias=False)
    self.transformer.wte.weight = self.lm_head.weight # weight tying

  def forward(self, idx):
    x = self.wte(idx)
    all_previous_x = [x] # This stores all the intermediary representations
    for i in range(self.config.n_blocks):
      x = self.blocks[i](x)
```

```
18        all_previous_x.append(x)
19        if (i+1) % self.config.dwa_period == 0: # Use DWA every dwa_period
20            all_previous_x_dilated = [x_ for j, x_ in enumerate(all_previous_x) if
          (len(all_previous_x)-1-j) % self.config.dilation == 0]
21            x = self.dwa_modules[i](all_previous_x_dilated) # Computing the weighted average with
          dilation
22        x = self.ln_f(x)
23        logits = self.lm_head(x)
24        return logits
```

## A.2  More Optimized Pytorch implementation

The point of this more optimized implementation is to remove the for loops of the naive implementation, and instead use tensor operations. To achieve this, we need to create a tensor containing the previous representations instead of a list. Given the structure of this tensor which will simply accumulate all the past representation during the forward pass, we would want to pre-allocate an accumulator and update this accumulator with a new representation after each block. However, this implies using in-place operations that conflict with Pytorch's autograd. We find a workaround by defining the InPlaceSetSlice class. We implement a helper module DWAModules built on top of this new class:

Listing 3: Content of the denseformer python package

```
1  import torch
2
3
4  class InPlaceSetSlice(torch.autograd.Function):
5
6    @staticmethod
7    def forward(ctx, full_tensor, last_slice, x_idx, x_val):
8      full_tensor[x_idx] = x_val
9      ctx.x_idx = x_idx
10     ret = torch.Tensor().to(full_tensor.device)
11     ret.set_(full_tensor[:x_idx + 1])
12     return ret
13
14   @staticmethod
15   def backward(ctx, grad_out):
16     if ctx.x_idx == 0:
17       return None, None, None, grad_out[ctx.x_idx]
18     else:
19       return None, grad_out[:ctx.x_idx], None, grad_out[ctx.x_idx]
20
21
22 def apply_inplace_set(x_acc, x_idx, x_val):
23   full_tensor, last_slice = x_acc
24   new_slice = InPlaceSetSlice.apply(full_tensor, last_slice, x_idx, x_val)
25   return full_tensor, new_slice
26
27
28 class DWAModules(torch.nn.Module):
29
30   def __init__(self, n_blocks, dilation=1, period=1):
31     super().__init__()
32     self.n_blocks = n_blocks
33     self.dilation = dilation
34     self.period = period
35     self.alphas = torch.nn.ModuleList([torch.nn.Linear((i+1+dilation)//dilation, 1, bias=False) if
       (i+1)%period == 0 else None for i in range(n_blocks)])
36     self.accumulators = None
37     self._init_weights()
38
39   def _init_weights(self):
40     for module in self.alphas:
41       if module is not None:
42         module.weight.data.zero_()
43         module.weight.data[0, -1] = 1.
44
45   def init_accumulators(self, x):
46     x_accs = []
47     for i in range(self.dilation):
48       current_group_size = (self.n_blocks + 1) // self.dilation
49       if i < (self.n_blocks + 1) % self.dilation:
50         current_group_size += 1
51       x_accs.append((torch.zeros((current_group_size, *x.shape), device=x.device, dtype=x.dtype),
       None))
52     x_accs[0] = apply_inplace_set(x_accs[0], 0, x)
53     self.accumulators = x_accs
54
55   def forward(self, x, block_idx):
56     assert self.accumulators is not None, "`init_accumulators(x)` needs to be called first"
57     self.accumulators[(block_idx+1) % self.dilation] = apply_inplace_set(
58       self.accumulators[(block_idx+1) % self.dilation],
59       (block_idx+1)//self.dilation,
60       x
61     )
62     if (block_idx+1) % self.period == 0:
63       x = torch.tensordot(self.alphas[block_idx].weight.view(-1),
64         self.accumulators[(block_idx+1)%self.dilation][1], dims=1)
65     return x
```

denseformer **python package.** The above module is available in the `denseformer` package which can be installed through the following link: https://github.com/epfml/DenseFormer. It provides the `DWAModules` class which orchestrates all the DWA logic given a number of blocks, a dilation factor, and a DWA period. After installing the package, a Transformer can be turned into a DenseFormer in three simple steps:

Listing 4: Faster $k$x$p$-DenseFormer implementation using the `denseformer` package.

```python
import torch
from denseformer import DWAModules

class DenseFormer(torch.nn.Module):

  def __init__(self, config):
    super().__init__()
    self.config = config
    self.dwa_modules = DWAModules(config.n_blocks, config.dilation, config.dwa_period) # Step 1
    self.wte = torch.nn.Embedding(config.vocab_size, config.n_embd)
    self.blocks = torch.nn.ModuleList([Block(config) for _ in range(config.n_blocks)])
    self.ln_f = LayerNorm(config.n_embd, bias=config.bias)
    self.lm_head = torch.nn.Linear(config.n_embd, config.vocab_size, bias=False)
    self.transformer.wte.weight = self.lm_head.weight

  def forward(self, idx):
    x = self.wte(idx)
    self.dwa_modules.init_accumulators(x) # Step 2
    for i in range(self.config.n_blocks):
      x = self.blocks[i](x)
      x = self.dwa_modules(x, block_idx=i) # Step 3
    x = self.ln_f(x)
    logits = self.lm_head(x)
    return logits
```

## A.3 Hyperparameters and Hardware Used

In all our experiments we trained on A100-80GB GPUs. In most of our experiments we trained using data-parallelism, distributing the batches over up to $4$ GPUs.

All of our models share the following hyperparameters:

- Max learning rate of $0.001$. For different depth, we tested values in $\{0.0001, 0.0003, 0.0005, 0.0007, 0.001, 0.002\}$ and found $0.001$ to be consistently better.
- 2000 warmup steps, chosen between $\{1000, 2000\}$
- A weight decay of $0.1$
- AdamW hyperparameters $\beta_1 = 0.9$ and $\beta_2 = 0.95$
- We use the GPT-2 tokenizer with a vocab size of $50304$
- 12 attention heads of $64$ dimensions
- An embedding size of $768$

# B Additional Results

## B.1 Information Flow

**Small $\alpha$ weights matter.** In the visualized weight matrix of Fig. 4, most of the weights on the inter-block connections are small. This observation raises the question as to whether it is possible to drop most of these connections or not. To answer this question, we plot the perplexity after dropping a portion of the smallest DWA weights and report the results in Fig. 6. It can be seen that even though a large portion of weights in Fig. 4 are small, dropping beyond 15% of the weights leads to a significant increase in perplexity. Therefore, even though these weights are small, they seemingly play an important role in predicting the next token.

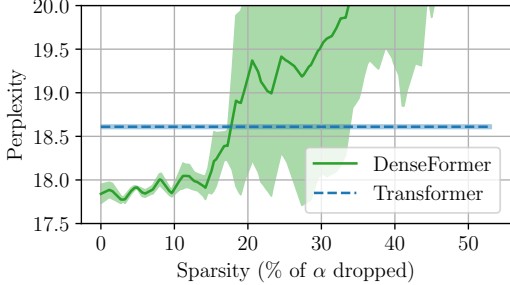

Figure 6: **Performance after dropping small DWA weights.** The figure shows the performance when the model is trained with no sparsity induced (dilation and period of 1) and the DWA weights are later sparsified based on their magnitude at inference. One can see that the perplexity quickly explodes after only sparsifying 15% of the weights. This observation suggests that even though many of the DWA weights are small (as can be seen in Fig. 4) they still play an important role in the output of the model.

**Visualizing $\alpha$s with periodicity.** Similarly to Fig. 4, we show in Fig. 7 the DWA weights for 4x3, 4x4, and 4x5-DenseFormers. We observe patterns similar to the 1x1-DenseFormer in Fig. 4 but at a lower resolution.

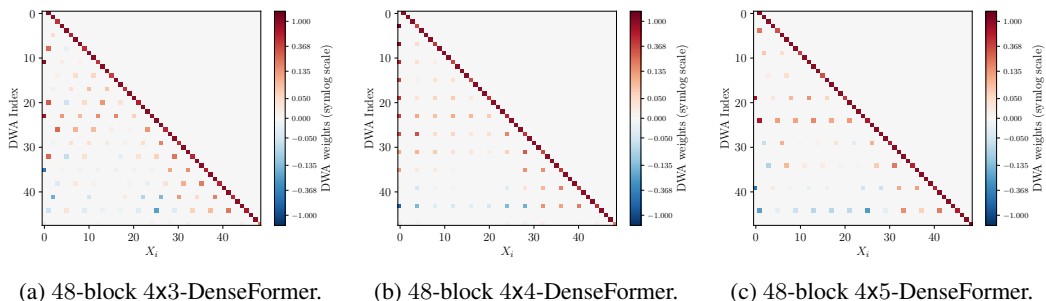

(a) 48-block 4x3-DenseFormer.  (b) 48-block 4x4-DenseFormer.  (c) 48-block 4x5-DenseFormer.

Figure 7: **Visualization of DWA Learned Weights.** Each row shows the weights $\alpha$ learned by a DWA module at a given depth. Those patterns are very consistent with the ones learned by a 1x1-DenseFormer, as seen in Fig. 4.

**Correlation with input embeddings at** 72 **blocks.** As in Fig. 5, we analyze the cosine similarity between the output of each DWA module and the initial embedding vectors for models of 72 blocks. The results in Fig. 8 are very consistent with those obtained with shallower models (Fig. 5).

**Evolution of DWA weights during training.** In Fig. 9, we plot the DWA weights of a 48 block DenseFormer during training. We observe how the pattern is learned relatively fast, within the first 5000 iterations.

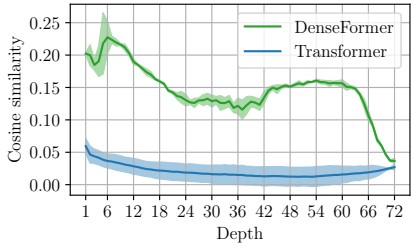

Figure 8: **Cosine similarity between the output of each DWA module and the initial embedding vectors.** The results are averaged over three seeds, for DenseFormer models with 72 blocks and no dilation. The model initially maintains a high correlation with the output of each DWA modules, but reduces that correlation towards later layers. Intuitively, we can hypothesize that this is the stage where the model is preparing to output the next token.

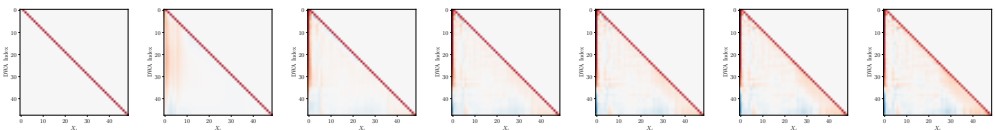

(a) Step 0    (b) Step 1000   (c) Step 2000   (d) Step 3000   (e) Step 4000   (f) Step 5000   (g) Step 6000

Figure 9: **Rapid convergence of DWA weights during training.** The DWA weights are rapidly converging to their final pattern. After 5000 iterations, the weight pattern already looks very similar to the one in Fig. 4.

## B.2    Analysis of Dilation and DWA Period

**More detailed analysis of dilation.** For 48 block models, we study the impact of varying the dilation factor $k$, we do not vary the DWA period which is set to 1. The results of this experiment are in Fig. 10. We observe how small dilation coefficients do not significantly deteriorate the perplexity yet increase the inference speed.

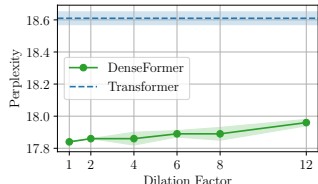 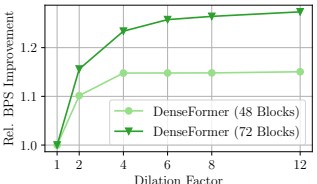

(a) Perplexity of 48 block $k$x1-DenseFormer on OWT2

(b) Relative speed improvement over dilation factor 1

Figure 10: **Effect of the Dilation Factor $k$ on Speed and Performance.** Part **(a)** shows the degradation in perplexity as we increase the dilation factor of $k$x1-DenseFormer models. A noticeable drop in performance occurs for larger dilation factors, e.g. after $k = 4$. However, surprisingly, 12-Dilated DenseFormer still outperforms the Transformer baseline. As shown in **(b)**, while the perplexity is not so impacted by dilation, the inference speed is significantly improved. Interestingly, the speed gain also plateaus for larger values of $k$, e.g. roughly $k = 4$ for 48 blocks. The gain increases with the depth of the DenseFormer, and the plateau threshold occurs later for deeper models.

**More detailed analysis of the DWA period.** For 48 block models, we study the impact of varying the DWA period $p$. We do not vary the dilation which is set to 4. In Fig. 11, we observe the impact of increasing $p$ on the perplexity. Interestingly, the perplexity profile is non-monotonic in $p$, which exposes the interplay between $k$, $p$, and the depth of the model. Moreover, increasing the DWA period further increases the inference speed over increasing the dilation.

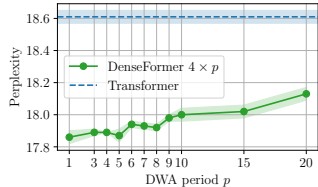
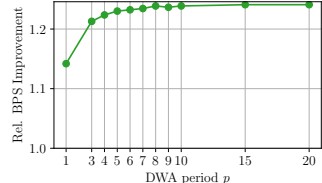

(a) Perplexity of 48 block 4x$p$-DenseFormer on OWT2

(b) Relative speed improvement over 1x1-DenseFormer

Figure 11: **Effect of the DWA period $p$ on Speed and Performance.** Part **(a)** shows the degradation in perplexity as we increase the DWA period of 4x$p$-DenseFormer models. Surprisingly, a 4x20-DenseFormer still outperforms the Transformer baseline. As shown in **(b)**, while the perplexity is not so impacted, the inference speed is significantly improved.

## B.3    PG-19 Experiments

**PG-19 experiments.** Tab. 4 shows the performance of DenseFormer against standard Transformer on PG-19, which consists of a large collection of full-length books from Project Gutenberg [24]. We trained both architectures for $48k$ steps and used a batch size of $128$ instead of $400$. All the other training parameters are kept the same as for OWT2 experiments. On this dataset, we can clearly see the superior performance of DenseFormer.

Table 4: **Comparison on PG-19.** Comparing DenseFormers and Transformers on the PG19 dataset. The results show similar improvements as the ones observed on the OpenWebText2 dataset. This demonstrates the generality of our results. Those results were obtained using a batch size of $128$.

| Model | Depth | Perplexity |
|---|---|---|
| Transformer | 24 | 20.13 |
| 1x1-DenseFormer | 24 | 19.60 |
| Transformer | 48 | 18.94 |
| 1x1-DenseFormer | 48 | **18.43** |
| Transformer | 72 | 18.44 |

## B.4    Delaying the Training of DWA Weights

In this section, we study what would happen if we started training the DWA weights at different training iterations. As seen in Fig. 9, the DWA weights are rapidly converging to their final values within the first 5000 iterations. Moreover, the initialization of the DWA weights corresponds to the same flow of information as in a normal transformer. This raises the question of whether training the DWA weights during the first training iterations is important, or whether a pre-trained model would still gain from adding the DWA weights later. To answer this question we experiment with training the DWA-weights after $N$ iterations. We do not modify the learning rate scheduler or any hyperparameter besides $N$. Results in Tab. 5 show a diminishing return as $N$ increases. It seems important to tune the DWA weights from the beginning. A possible hypothesis could be that the iterates commit to a valley in the loss landscape relatively early during training. Once deciding to go to the valley where DWA weights are not used, it is difficult to recover and ultimately benefit from newly added DWA weights. We believe this phenomenon could be mitigated using a better learning rate scheduler. We leave this investigation as future work.

## B.5    Rank Analysis

In this section, we compare the ranks of matrices learned using DenseFormer and Transformer architectures. Our main result in Fig. 12 is that there is no significant difference in rank between the two approaches.

## B.6    Experiments with a batch size of $128$

In this section, we revisit experiments from the main paper but use a small batch size of $128$ instead of $400$ during training.

Table 5: **Start training the DWA weights after** $N$ **iterations.** At initialization, a DenseFormer is the same as a Transformer. We experiment with tuning the DWA weights only after $N$ iterations. This means the model is trained as a Transformer for $N$ iterations, and as a DenseFormer from $N$ to 40k iterations.

| Model | N | Perplexity |
|---|---|---|
| Baseline Transformer | - | 18.61 (0.02) |
| 4x5-DenseFormer | 0 | **17.87 (0.02)** |
| 4x5-DenseFormer | 1k | 17.99 |
| 4x5-DenseFormer | 2k | 18.07 |
| 4x5-DenseFormer | 4k | 18.13 |
| 4x5-DenseFormer | 6k | 18.17 |
| 4x5-DenseFormer | 10k | 18.23 |
| 4x5-DenseFormer | 20k | 18.33 |
| 4x5-DenseFormer | 30k | 18.40 |

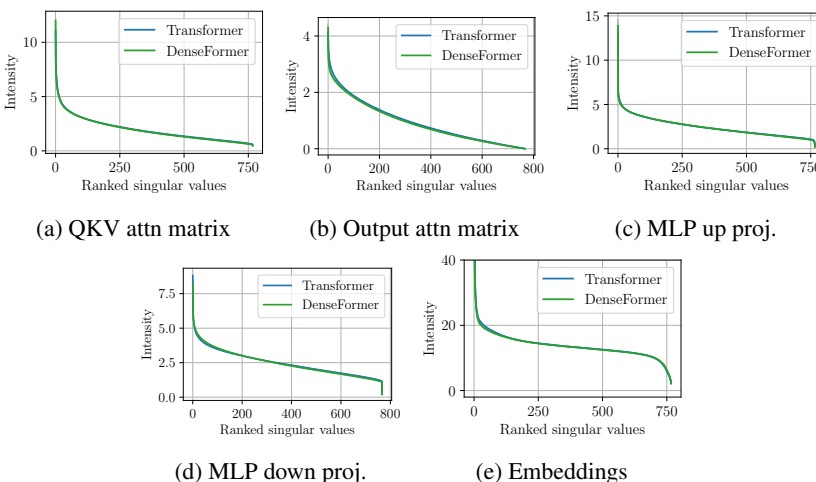

(a) QKV attn matrix     (b) Output attn matrix     (c) MLP up proj.

(d) MLP down proj.     (e) Embeddings

Figure 12: **Ranked singular values averaged across blocks.** For 48 block models, we average the singular values for each matrix across blocks (except for the embedding matrix). We observe no significant differences between Transformers and DenseFormers. Results are averaged over 3 seeds.

**Speed and performance trade-off.** In Fig. 13 we show the trade-off between inference speed and perplexity for different numbers of blocks. Similarly to Fig. 2b, DenseFormers reach a better perplexity than much deeper Transformer models. Interestingly, the perplexity gap is larger than when using larger batches (compared to a batch size of 400 used in Fig. 2b). A 48 block DenseFormer is performing on par with a 90 block Transformer. This might indicate that the DenseFormer is more robust to large gradient noise compared to Transformers. DenseFormers reach better trade-offs in terms of inference speed and perplexity. Those results are expected to improve if we were to train a 4x5-DenseFormer instead of a 4x1-DenseFormer. Detailed results can be seen in Tab. 6.

**Results with other sparse patterns.** In Tab. 7 we reproduce the experiments of Tab. 3 but with a batch size of 128. Similar conclusions follow.

**Visualizing the DWA weights (trained with a batch size of** 128**).** In Fig. 14 we plot the DWA weights obtained when training with a batch size of 128. The learned patterns are very consistent with the ones of Fig. 4 obtained with a larger batch size.

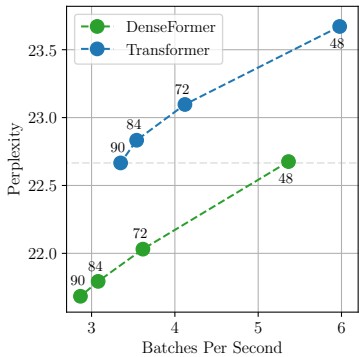

Figure 13: **Speed and performance trade-off.** Comparison of speed and performance trade-off between the standard Transformer architecture and 4x1-DenseFormer. The number of blocks in each architecture is reported next to the data-point. All DenseFormer models on this plot use a dilation factor of 4. **Comparing perplexities:** Considering only the perplexity (y-axis), a 48 layer DenseFormer strikingly outperforms much deeper Transformer baselines. **Comparing trade-offs:** A 48 layer 4-Dilated DenseFormer matches the better perplexity of a 90 layer Transformer while being 1.6× faster at inference.

Table 6: **Performance of DenseFormer and the standard architecture of different sizes on OpenWebText2 dataset.** Using a batch size of 128, and a DWA period of 1. DenseFormer clearly outperforms a standard architecture of the same depth as well as standard architecture with the same inference speed. While sometimes a deeper model with the standard architecture can match the performance of a shallower DenseFormer, inference using the shallow DenseFormer remains much faster.

| Model | Dilation | Depth | Parameters # (M) | Perplexity | Inference BPS |
|---|---|---|---|---|---|
| Transformer | - | 48 | 378.45 | 23.67 (0.09) | 5.98 (0.00) |
| Skips With Gains | - | 48 | 378.45 | 23.78 (0.19) | 5.72 (0.01) |
| | 1 | 48 | 378.45 | 22.61 (0.05) | 4.67 (0.00) |
| DenseFormer | 2 | 48 | 378.45 | **22.60 (0.04)** | 5.15 (0.01) |
| | 4 | 48 | 378.45 | 22.68 (0.06) | **5.36 (0.00)** |
| Transformer | - | 64 | 491.72 | 23.21 (0.07) | 4.59 (0.00) |
| | - | 72 | 548.35 | 23.10 (0.02) | 4.12 (0.00) |
| | 1 | 72 | 548.36 | **21.81 (0.00)** | 2.93 (0.00) |
| DenseFormer | 2 | 72 | 548.35 | 21.92 (0.04) | 3.39 (0.00) |
| | 4 | 72 | 548.35 | 22.03 (0.04) | **3.62 (0.00)** |
| Transformer | - | 84 | 633.31 | 22.84 (0.07) | 3.56 (0.00) |
| | - | 90 | 675.78 | 22.67 (0.04) | 3.35 (0.00) |

Table 7: **Alternative DWA Sparsity Patterns.** We compare 48 block architectures with different sparsity patterns. In Last K, DWA can only access the output of the last $k$ blocks as well as the embedding vectors. Connect to Last includes only a single DWA module in the architecture placed after the last layer, i.e. connecting every block to the last block. Neither of these patterns allows achieving the same perplexity boost as the original DenseFormer. Even with a dilation of 12, which implies a sparsity of 92%, the Denseformer models outperform other sparsity patterns, which signifies the importance of pairwise inter-block connections.

| Sparsity Pattern | Perplexity | Sparsity |
|---|---|---|
| Baseline Transformer | 23.67 (0.09) | - |
| 12x1-DenseFormer | **22.91 (0.06)** | 92% |
| Last K ($K = 4$) | 23.23 (0.07) | 84% |
| Connect to Last | 23.45 (0.05) | 96% |

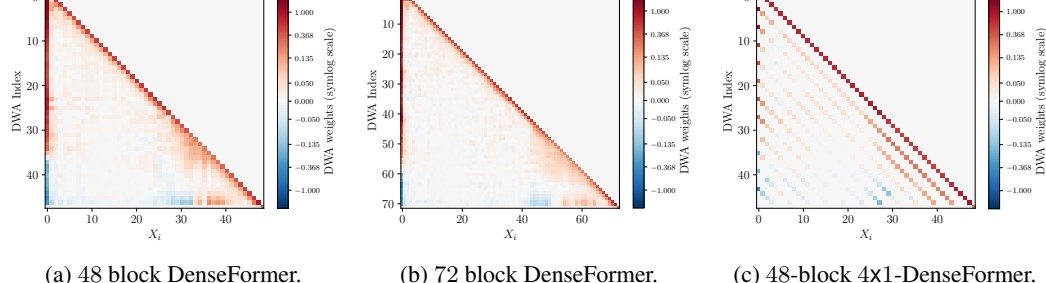

(a) 48 block DenseFormer.  (b) 72 block DenseFormer.  (c) 48-block 4x1-DenseFormer.

Figure 14: **Visualization of DWA Learned Weights.** Each row shows the weights $\alpha$ learned by a DWA module at a given depth. While the heatmaps are averaged across 3 runs with different seeds, those patterns are very consistent across seeds. In **(a) and (b)**, strikingly similar patterns can be observed in both 48 and 72 layer DenseFormers. In **(c)**, we show the learned weights for a 48 block DenseFormer trained with a dilation of 4. Despite the sparsity, we still observe a very similar pattern to those learned by the non-dilated models.

## B.7 Comparison with ELC-BERT

In this section we compare our work with the concurrent work of Charpentier & Samuel [2]. We implement the different variants they suggest and show the results we obtain in Fig. 15. We observe that while their architecture is more data efficient in the low data regime, DenseFormers catch-up quickly (after about 1.2k iterations). In the non low data regime, ELC-Bert performs similarly as the baseline, while DenseFormer is consistently better.

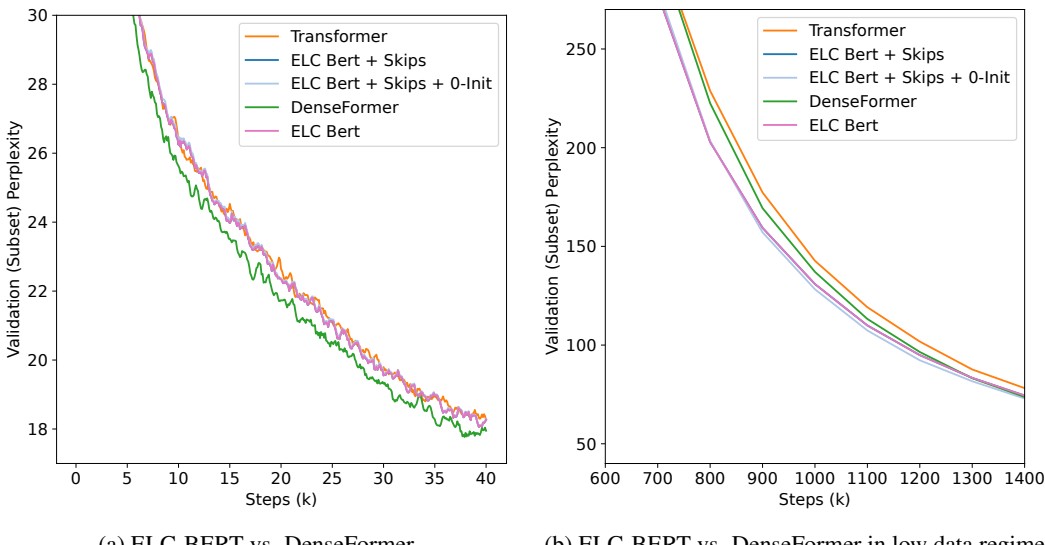

(a) ELC-BERT vs. DenseFormer  (b) ELC-BERT vs. DenseFormer in low data regime

Figure 15: **Comparison with ELC-Bert.** Part **(a)** Shows that ELC-Bert does not improve upon the baseline, while DenseFormer is consistently better. In **(b)**, we observe that ELC-Bert is better in low data-regime, which is the setting for which it was developed. Yet the DenseFormer is catching up fast and outperforms ELC-Bert after approx. 1.2k iterations.

## B.8 Effect of varying the width

**Information capacity bottleneck.** We can postulate that the larger the width of our model, the easier it is to pass information sequentially from layer to layer. Therefore, the wider the network, the less we expect to gain from adding inter-layer residual connections. We experiment with varying the width of 24 block models, using the following values: $\{384, 768, 1536\}$. Results are summarized in Table 8. We also train a transformer model with a larger hidden size of $1664$. For the larger widths of $1536$ and $1664$, we tuned the learning rate in $\{0.001, 0.0005, 0.0003\}$ and found $0.0005$ to be best. We use a batch size of $128$, a learning rate of $0.001$, a sequence length of $256$, and train for $40k$ iterations. We observe—as expected—the gap becomes smaller for wider models, yet does not disappear. A DenseFormer with a smaller hidden size of $1536$ performs better than a Transformer with a hidden size of $1664$. In general, as the complexity of the task increases, followed by the need for more capacity, it is more compute and memory efficient to use a DenseFormer instead of a Transformer. This is especially relevant when we think of deploying LLM on devices with hardware constraints.

Table 8: **Effect of varying the width.** .

| Width | Transformer PPL | DenseFormer PPL | Parameters |
|-------|-----------------|-----------------|------------|
| 384   | 31.039          | 30.120          | 62M        |
| 768   | 25.020          | 24.631          | 209M       |
| 1536  | 21.597          | 21.279          | 757M       |
| 1664  | 21.313          | -               | 881M       |

