# OpenReview forum: "DenseFormer: Enhancing Information Flow in Transformers via Depth Weighted Averaging"
_NeurIPS.cc/2024/Conference — NeurIPS 2024 poster_

### Official Review · Reviewer_EGgo · 2024-07-07

**Soundness:** 3
**Presentation:** 4
**Contribution:** 3
**Rating:** 7
**Confidence:** 4

**Summary:**

The authors propose to use a weighted average of previous layer outputs as an input for the successive layers. The weights are learned parameters and can also take negative values. The weighting module can be coarse: it is enough to insert it only in some layers, and it is enough to attend to a subset of previous layers, which significantly lowers the overhead resulting from the weighting. The authors show perplexity gains on language modelling on OpenWebText2. They analyze the learned weights, and uncover interesting patterns.

**Strengths:**

The method is easy to use, improves perplexity, and shows interesting and consistent weight patterns. The overhead is minimal, less than for depth-wise attention, and the perplexity in a large data regime is better than that of the concurrent DenseNets.

**Weaknesses:**

The evaluation could be stronger: It would be nice to evaluate the zero-shot performance on some downstream tasks (e.g., BLiMP, CBT, PIQA). This should be easy to implement and does not require additional training.

The evaluated transformers are narrow ($d_{model}=768$) and very deep (48 layers for the 378 and 72 for the 548M parameter models). It is unclear how the gains would transfer to a more traditional setup (e.g a 350M GPT-3 has 24 layers). The effects might be smaller with fewer layers if the main role of the DWA module is to provide additional shortcuts for the gradients. The unusually narrow $d_{model}$ can provide an additional advantage for DWA because there is less space to store everything in the residual directly, and the shortcuts can help with that. There are two ways to improve this: either show the gains with a more standard aspect ratio or show that the narrow+deep DWA model is better than a more traditionally shaped DWA model (both of these should be paramer-matched).

**Questions:**

In line 133, the authors write, "the outputs are stored in KV cache to facilitate decoding.". Depending on the implementation, this is not what is typically done. To avoid recomputation, the K and V values are stored instead of the residual. However, I agree with the authors that this is not an issue since only the state of a single column has to be stored, which is typically negligible.

**Limitations:**

Already discussed in the weaknesses section. The authors should add some discussion on the limitations of the evaluation.

---

> ### Author Rebuttal · Authors · 2024-08-07
>
> We thank the reviewer for taking the time to review our work.
>
> 1. **On the KV cache comment (line 133).** We thank the reviewer for bringing this to our attention and agree the statement line 133 is incomplete. As the reviewer rightfully pointed out, while we do not need to store the past layers’ outputs, a small price has to be paid to store the intermediary states for the current token. This cost is small and does not grow with the sequence length. We will make this clear in our next revision.
> 2. **On the width used in our experiments.** We agree that our DenseFormer architecture helps to solve the capacity bottleneck of the residual stream typically present in transformers. As such, a narrow DenseFormer can match the performances of a wider Transformer. This is an important advantage as increasing the width also increases the training and inference costs in terms of both compute and memory. To investigate the impact of the model’s width on the perplexity, we trained multiple $24$ layer models with different widths of $384$, $768$, and $1536$. The results are summarized in the following table:
>
>     | Width      | Transformer | DenseFormer | Parameters  |
>     | -----------| ----------- | ----------- | ----------- |
>     | 384        |    31.039  |    **30.120**   |    62M      |
>     | 768        |    25.020   |    **24.631**   |    209M     |
>     | 1536       |    21.597   |    **21.279**   |    757M     |
>     | 1664       |    21.313   |       -     |    881M     |
>
>     We observe that the gap—while smaller for wider models—does not disappear. Moreover, a $1536$ wide DenseFormer performs better than a $1664$ wide Transformer, despite having $124$M fewer parameters. In general, as the complexity of the task increases, followed by the need for more capacity, it is more compute and memory efficient to use a DenseFormer instead of a Transformer. This is especially relevant when we think of deploying LLM on devices with hardware constraints.
>
>
>     Additionally, other hypotheses exist that could explain the advantage of DenseFormers over Transformers. For instance, we mention in our work how the DWA weights can learn to subtract earlier representations from later representations, which potentially helps to disentangle between processing the current token, and predicting the next token. Evidence for this can be seen in Fig.4 and Fig.5.
>
> We hope the above comments address your concerns and that you will consider raising your score.

---

> > ### Comment · Reviewer_EGgo · 2024-08-10
> >
> > I would like to thank the authors for the time and effort invested in their rebuttal.
> >
> > I appreciate their new experiment investigating more standard model shapes and am glad to see that the gains still hold. Thus, I am increasing the score.

---

> > > ### Author Response · Authors · 2024-08-12
> > >
> > > We are pleased that the reviewer found our response helpful and sincerely thank them for their decision to increase their score.

---

### Official Review · Reviewer_1YUF · 2024-07-11

**Soundness:** 3
**Presentation:** 4
**Contribution:** 4
**Rating:** 7
**Confidence:** 4

**Summary:**

The work introduces DenseFormer, a variant of Transformer architecture using special connections between blocks. Each transformer block in DenseFormer may look at a weighted average of all previous blocks' outputs (instead of the simple output of the previous block). Authors run extensive experiments showing the performance improvement of such a design and show that this simple change beats the baseline model on the Pareto-frontier of model quality and speed. Authors also run good ablations and showcase different variants of DenseFormer.

**Strengths:**

1. **The idea.** The idea behind the work is simple, in a very positive way, with a good story. The implementation is straightforward (even if optimizations are useful), the methods aren't costly during training, and what costs it incurs it makes up for in performance. Moreover, I believe the technique introduced could not only improve Transformer-based language models but Transformers in different modalities as well, and even other kinds of residual networks apart from Transformer - not only "classical" CNN-based ResNets, but also new architectures like Mamba.
2. **Experimental setup.** Experiments are extensive, and results show consistent improvement. Every metric that I'd want to see is present and measured, like inference throughput, training speed, and comparison to models equal in inference/training time (both through an increase in size and through an increase in training iterations).
3. **Improvements on Pareto-frontier.** The significant Improvements achieved with DenseFormer compared to the "vanilla" Transformer should interest both the research community and industry applications.

What is more, **the authors provide excellent, easy-to-understand code**, so even when I had doubts about how things were implemented, I could easily check the source code; thank you so much.

**Weaknesses:**

**Non-standard model shape (issue essentially resolved during the rebuttal).** By a model shape, I mean the relative dmodel and nlayers. I am extremely worried that results would hold up on standard model shapes. The work uses a dmodel of 768, with the smallest models being 48 layers deep and the biggest model being 90 layers. This is extremely deep and thin.

For reference I will use GPT3 model shapes, as they are reasonably tuned and still used by the community today ( see Table 2.1 on page 8 of https://arxiv.org/pdf/2005.14165 ). The only model with a dmodel of 768 has just 12 layers, 4x shallower than **the shallowest** model used by the authors. This ratio 768/12 was also used by BERT-base. The common ratio used by the community, shared by GPT3 6.7B, llama7B, Mistral, is 4096 dmodel with 32 layers - still shallower than any models used by the authors, and 6x as wide! Chinchilla 80B and LLama 80B use 8192/80, with depth comparable to the authors' models. Only when we get to "the GPT3", 175B, we finally get to the model deeper than any of the authors' models, but with a whopping 12288 dmodel.

The above issue concerns me because one possible interpretation of DenseFormer improvements is essentially increasing the information throughput/capacity of the residual stream, as the layers can communicate directly without using the limited residual stream. With those hyperparameters artificially bottlenecking the throughput of the baseline model's residual stream (compared to the optimal hyperparameters), every technique that effectively circumvents the bottleneck gains an unfair advantage, and the results may not reflect the improvement in the real-world models.

I am unsure how big of an impact this nonstandard architecture has (maybe it doesn't have any impact?), but it leads me to trust the results less. While I trust that running wider models isn't feasible with computational constraints, I would suggest the authors run models of "standard" shape, especially when it comes to the baselines. For example, seeing the results of a smaller model, like 768/12 (and/or less than 12 layers while keeping the dmodel of 768), would showcase the impact of changing the width/depth ratio of the model on the relative improvement achieved by DenseFormer. Those experiments would also be relatively cheap (I'm not suggesting authors pre-train 7B models, of course).

The saving grace for now is that the code exists and everything is easy to reproduce by the community, so I'm leaning accept even with this issue standing. However, running those experiments would be perfect.

**Learning rate, hyperparameter tuning.** Those are less important than the depth/width ratio, but still important. Learning rate (and other parameters, if tuned). How was it chosen? Was it tuned for the baseline? Was it tuned at all? This, again, is important - especially for those deep models, which are generally harder to train and less stable.

**Minor**
1. Number of heads. In line 198, "We use models with 8 heads", while in line 602, "12 attention heads". One of those lines is incorrect (I assume 12 heads is used, though).
2. I needed to see the code to see the details about the residual connection inside the block and where the layer norm is (is Pre-LN used, in particular). Adding a "Block" implementation to the paper's naive implementation code would be nice to make it easier for future readers to check this. That said, the code is very nicely written.
3. It would be worth adding that the memory overhead (line 130) is only negligible if no activation checkpointing is used (and it is commonly used). Also, again, for the inference, the overhead is negligible if no optimizations are used to shrink the KV cache (and again, things like grouped attention are commonly used).

**Questions:**

See weaknesses. The primary concern is the architecture shape, and the secondary is hyperparameter tuning. With this primary issue resolved, this would be outstanding work.

Another question is, can we get a more detailed analysis of how exactly the technique differs from previous work, apart from being applied to Transformers? My understanding is that it has multiple novel parts that were adapted specifically, but more detailed comparison would be welcome.

**Limitations:**

Properly addressed, except for the architecture shape (see weaknesses).

---

> ### Author Rebuttal · Authors · 2024-08-07
>
> Dear Reviewer,
>
> Thank you for your careful consideration of our work and your valuable feedback. We provide the following comments to further clarify our contributions:
>
>
> 1. **On the non-standard model shapes.** As you have correctly pointed out, using DenseFormer “alleviates the information capacity bottleneck of the residual stream”. It is also true that increasing the hidden dimension is another possibility for resolving this bottleneck. Indeed, we conjecture that given a DenseFormer it is possible to build a standard model with a much larger hidden dimension and show that it can theoretically perform the same operations as the original DenseFormer. As such, it is possible that the gap becomes less pronounced as we increase the width. However, increasing the hidden dimension comes with significant overheads on the resources, leading to larger models and additional memory footprint (e.g. larger KV cache). To further strengthen our claims, we additionally ran experiments with a $24$ layer DenseFormer and compared it at different widths of $384$, $768$, and $1536$ with a standard Transformer. The perplexities for those models are presented in the following table:
>
>     | Width      | Transformer | DenseFormer | Parameters  |
>     | -----------| ----------- | ----------- | ----------- |
>     | 384        |    31.039  |    **30.120**   |    62M      |
>     | 768        |    25.020   |    **24.631**   |    209M     |
>     | 1536       |    21.597   |    **21.279**   |    757M     |
>     | 1664       |    21.313   |       -     |    881M     |
>
>     For the larger widths of $1536$ and $1664$, we tuned the learning rate in $\{0.001,0.0005,0.0003\}$ and found $0.0005$ to be best. We can see that the gap between the two architectures still persists. We also validate the above conjecture by measuring the perplexity of a transformer model with a larger hidden size of $1664$, which is worse than a DenseFormer with a smaller hidden size of $1536$. This, despite a $124$M parameters difference. Furthermore, given the above discussion regarding the trade-off between capacity, width, and using DenseFormer, we expect the gap to widen when moving to more demanding settings which require additional capacity (e.g. longer sequence lengths, or training on more tokens).
> 2. Moreover, while current architectures are using larger hidden dimensions, this can be because until now, increasing the hidden dimension has been the only avenue to address the said information bottleneck. In our work, we point out that using DWA additionally allows operations such as subtracting the first layer’s output which we have demonstrated is actually learned by the model.
> 3. **On the learning rate and hyperparameters tuning.** We tried our best to tune the hyperparameters for the baseline models at different model depths given the available resources. We then used the same settings when training DenseFormers. We tested learning rate values in $\{0.0001,0.0003,0.0005,0.0007,0.001,0.002\}$, and found $0.001$ to be systematically better. We also tuned the number of warmup steps for a $48$-layer standard Transformer, and used the best values found in all subsequent experiments. We will include these clarifications in our next revision.
> 4. **On the memory overhead.** We agree that when activation checkpointing is used, the overhead might become more noticeable at training and thank the reviewer for pointing this out. We mention that at the decoding stage during inference, we do not need to store the intermediate outputs for already processed tokens. The overhead of storing the intermediate outputs for a single token remains negligible in comparison with the KV cache.  We will include these clarifications in the next revision of our paper.
> 5. **On the difference with relevant prior works.** The most relevant prior method might be DenseNet. DenseNet is composed of a succession of Dense blocks. Within each block, convolution operations are interleaved with concatenation operations, which concatenate the output of all the previous convolution blocks. Therefore, within a dense block, the size of the representation is increasing. The hidden size is reduced using transition blocks consisting of convolution and pooling. Compared to DenseNet, our approach is minimalist: we simply do a linear combination of past layer’s outputs. This removes the need for transition blocks, only adding a very small number of parameters. We will include this discussion in the related work section of our next revision and try making the distinction with prior work more clear.
>
> We hope the above comments address your concerns and that you would consider raising your score.

---

> ### Comment · Reviewer_1YUF · 2024-08-09
>
> I want to thank the authors for their response, and **I have raised my score (from 6 to 7)** - as I assume a reasonable response will be given to the remaining questions regarding the training setup (see below).
>
> **On model shapes.** Thank you for providing experimental results. I'm glad to see that you also included a pretty standard architecture 1536/24 (e.g., GPT-Large) and a wider range of model parameters (881M vs. 62M, so a 14x difference, instead of around 2x in the original work). While ideally, I would recommend going for more standard shapes of models, especially with a reasonable width-to-depth ratio, I agree the presented results show that DenseFormer should show gain in practice as well.
>
> Those experiments show considerably worse perplexity for a given model size than experiments in the paper. What was their training length? I assume this must have been the critical difference—or was it something else? It looks like 16k steps or so instead of 40k. Or something else was changed as well?
>
> **On other things.** The rebuttal answered my questions. Just to confirm - the learning rates tuning was done separately for the Transformer baseline and separately for DenseFormer. Am I understanding this correctly?
>
> **Change in score from 6 to 7.** Assuming that the above results (with an explanation of differences in the training setup or explanation of worse perplexity in those wider models), a brief note of hyperparameter tuning like the one in the rebuttal, and the expanded comparison to DenseNet like the one in the rebuttal, will be included in the paper or the appendix for the benefit of readers - I raise my score (from 6 to 7), as I'd like this work to be seen at NeurIPS.

---

> > ### Author Response · Authors · 2024-08-12
> >
> > We thank the reviewer for increasing their score. We will add the new width experiment to our next revision, along with a description of the hyperparameters training procedure.
> >
> > * **Experimental details for the width experiment.** As we changed the width of the models and reduced the depth, we had to re-tune the learning rate. For both Transformers and DenseFormers alike, we ended up using a learning rate of $0.001$ for the widths of $384$ and $768$, and a learning rate of $0.0005$ for models of width $1536$. In the interest of time, we decided to reduce the batch size from $400$ to $128$. As a result, the models are trained on less tokens, which explain the gap in perplexity. Remaining hyperparameters are the same as for our other experiments, i.e. we use $40k$ iterations and a sequence length of $256$.
> >
> > * **Concerning the tuning of the learning rate for DenseFormer and Transformer models.** We extensively tuned the learning rates for our Transformer models for all depths. We then mostly applied those learning rates to our DenseFormer models. On a small set of experiments, we verified that the added DWA modules do not seem to be affecting the optimal learning rate value.
> >
> > We thank again the reviewer for his valuable feedback, and stay at his disposal in case any further clarification is needed.

---

### Official Review · Reviewer_VTZd · 2024-07-12

**Soundness:** 3
**Presentation:** 4
**Contribution:** 4
**Rating:** 7
**Confidence:** 4

**Summary:**

The paper introduced DenseFormer -- a simple and effective architecture that boost transformer language model's performance by adding trainable residual connections to all the previous layers. The paper discusses intuitions behind the architecture (that it enhances information flow between earlier and latter layers) and design choices for quality-cost tradeoff. Empirical results on the language modeling tasks validates the effectiveness of the architecture. The paper also presents empirical observations of the residual weights which exhibit consistent patterns that might be tied to the effectiveness of the architecture.

**Strengths:**

Originality: while the idea of residual connection or trainable residual connection have been proposed, it is still surprising that a simple trainable combination of residual streams can be both effective and has consistent weight patterns. The design of this architecture and the findings of the effectiveness and weight pattern are novel.

Clarity: the paper presents the idea in a clear and easy-to-follow fashion. Various design choices are discussed regarding quality-cost trade-off; and important issues such as whether the architecture remains effective if introduced in latter training stages are discussed.

Significance: the simplicity and effectiveness of the method is quite appealing for consideration in practical language models.

**Weaknesses:**

A minor weakness is the robustness of the architecture across tasks: the paper focuses on the language modeling task, while transformers have been used for vision, speech and general time series data. A discussion on the robustness of Denseformer on other tasks would strength the paper.

**Questions:**

Regarding the memory consumption of DenseFormer, the paper mentioned that it has "Negligible Memory Overhead" since the previous layer's output are stored in the KV-cache. This seems to be at odd with the most common practice, in which the actual key and value vectors are stored in the KV-cache, not the layers' outputs. The reviewer assumes that the authors are referring to a difference way of caching -- caching the layers' output and re-compute key, value projections at inference time, which will be consistent with the negligible memory overhead claim, but introduces additional computation. A clarification on this point would be helpful.

---

> ### Author Rebuttal · Authors · 2024-08-07
>
> Dear Reviewer,
>
> Thank you for your consideration of our work and your valuable feedback. We make the following comments to provide further clarifications:
>
> 1. We completely agree on the ambiguity regarding the negligible memory overhead claim and thank you for bringing it to our attention. We point out that at the decoding stage during inference, we do not need to store the intermediate outputs for already processed tokens. The overhead of storing the intermediate outputs for a single token remains negligible in comparison with the KV cache. However, a more careful implementation is needed to avoid additional memory usage during the pre-fill stage as you suggested. We will clarify this in the revision of our paper.
> 2. While we expect our results to extend to other tasks as well, we decided to focus on text for this work to save on resources. We agree that extending the results to other tasks would be an excellent direction for future work.
>
> Thank you.

---

### Official Review · Reviewer_Ljsg · 2024-07-25

**Soundness:** 2
**Presentation:** 3
**Contribution:** 2
**Rating:** 4
**Confidence:** 4

**Summary:**

The paper proposes to construct current transformer block input by weighted average of all inputs from previous transformer blocks. The weights are static and learned during training process. To reduce computation complexity, the method comes with a dialated version controlled by modulo and division relations. Experiments are conducted to observe the changes in model size, train/test speed, PPL. Models are trained from scratch on OpenWebText2. Results show better PPL, or similar PPL with smaller/faster model.

**Strengths:**

the method is well defined, and experimental results are good

**Weaknesses:**

not a rigorous study regarding experiment comparison with related work, dialation mechanism design

**Questions:**

1. the abstract is a little misleading, L5 says "100B parameters", but experimetal models are at the level of millions of parameters.
2. the proposed method should consider a generalization into triangular/pyramid structure. what happens if there is DWA on top of DWA and again on top of DWA?
3. the sparsity design by modulo and division is rule based (with no intuition on hyperparameter setting), not learned from training. is it possible a simple regularization can be much better?
4. the whole design is meant for traning models from scratch. it becomes not useful given many well trained transformers are released.

---

> ### Author Rebuttal · Authors · 2024-08-07
>
> We thank the reviewer for taking the time to review our work. We hereby answer the reviewer’s questions in order:
>
> 1. The number of additional parameters required by DenseFormer increases quadratically with the model’s depth. We mentioned 100B parameter models as those large scale models are typically the deepest and represent a worst case scenario. This serves to emphasize that even in this worst case, the number of added parameters is negligible.
> 2. If we understand correctly, it is suggested to add DWA on top of other DWAs. Given that the current DenseFormer architecture is already mapping any block output to all future block inputs, adding additional connections on top would not make the model more expressive. Please let us know if we misunderstood your statement.
> 3. We opt for rule based sparsity for simplicity and to maintain hardware compatibility. Other types of sparsity do not easily yield speedups or memory savings and require modifying the loss, possibly introducing additional training overheads. Furthermore, as can be seen in Fig 3.a., the rule based sparsity is already doing quite well, allowing 4x5 DenseFormers to achieve a very close performance to 1x1 DenseFormers. The remaining gap seems too small to justify deploying adaptive sparsity techniques.
> 4. The reviewer mentions it is not relevant to do research on better transformer training given open-source models are being released. We strongly disagree with the reviewer on this statement. Just because pre-trained LLMs are open sourced does not mean it is not worth researching more efficient ways to train LLMs. The field is extremely fast-paced and new models are continuously trained and released. Our research---by making models more parameter efficient---increases the expected return from pre-training models, and therefore promotes the training of more models.
>
> We hope we have answered your concerns, in which case we would appreciate it if you could raise your score.

---

> ### Comment · Reviewer_Ljsg · 2024-08-08
>
> 1. parameter amount scale can greatly affect model performance. current experimental results are useful for million level models. however, it may be more useful, less useful, useless or possibly performance-hurting for billion level models. the hypothesis is inclined to being less useful or useless, as large models are harder and much more challenging to improve.
> 2. the question is looking to see the potential or the upper bound of performance improvements from using DWA. it is straightforward to test by recursively adding DWA on top of DWA to form a pyramid structure, which should not be an issue considering DWA parameter amount are quite small.
> 3. rule based sparsity design is easy to test and start. given sparsity design is an important part of the paper, it takes 2 sections (3.2 & 3.3) of method writing. it is expected that authors would naturally explore other simple design options, such as regularization based methods.
> 4. the proposed method seems to only require adding few parameters, but at the cost of training the entire model from scratch. in fact, such limitation is directly or indirectly related to above comment 1/2/3, because the huge training cost hinders many scientific explorations.
>
> Overall, it is challegening and expensive for authors or for the community to verify its effectiveness on billion level models, which are common in the community. to be clear, this is not a question/requirement on author's experimental equipment setup, but a question on the applicability, generalization, and usefulness (on billion level models) of the method.

---

> > ### Comment · Reviewer_1YUF · 2024-08-10
> >
> > I am responding to a reviewer as another reviewer. I agree with Reviewer Ljsg in some respects. For example, **(point 1)** I also think the mention of the 100B model in the abstract is misleading, and the same thing should be phrased differently w/o implying the method was tested on such models (replacing the sentence with, e.g., "adding less than a percent of total model params").
> >
> > I also partially agree with Reviewer Ljsg that **(point 4)** pretraining models are costly, and it is a clear limitation of the technique that it requires pretraining. For sure, I'd like to see experiments on Transformer-to-DenseFormer fine-tuning of the Llama or other models, probably as future work. With that said, I find the arguments about "the cost of training the entire model from scratch" or "the huge training cost hinders many scientific explorations" - **those arguments apply not only to this paper but to all papers on improving LLM pretraining** - so I would not hold that argument against authors. I'd like to see more research on more efficient and better pretraining, after all. I'd suggest to the authors, e.g., extrapolating the performance of the model to larger scales (like comparing scaling laws for DenseFormer vs for Transformer), but such an extrapolation also would require more experiments.
> >
> > Regarding **(point 2)**, it seems to me that adding DWA on top of DWA would not change the model's expressiveness. Reading Reviewer Ljsg's suggestion, I don't know how the proposed technique would be meaningfully different from what is already tested in the paper.
> >
> > Regarding **(point 3)**, while I agree with Reviewer Ljsg that regularization-based sparsity would be scientifically interesting, I also agree with the authors that the existing static sparsity pattern seems to capture the majority of the gains. Given this, it seems to me that any kind of regularization-based sparsity would be impractical because it introduces additional complexity both in terms of training (auxiliary losses) and in hardware/performance optimizations. The computational budget for such experiments could be better spent elsewhere (e.g., larger models or more experiments for determining scaling laws).

---

> > > ### Comment · Reviewer_EGgo · 2024-08-10
> > >
> > > I am also responding to reviewers Ljsg and 1YUF as a reviewer.
> > >
> > > Specifically, I strongly disagree with reviewer Ljsg's point 4. With such reasoning, we should stop all research that is not fine-tuning and prompting and hope that, somehow, all the remaining shortcomings of the models will be fixed by themselves. This is obviously the wrong direction. In my opinion, the most interesting research is the one that targets fundamental limitations of the models. These typically require retraining of the model. Maybe such research is not directly useful for very applied research but is the most useful for advancing the field.
> > >
> > > Regarding points 2 and 3, I think that the fact that high dilations and periods retrain almost all gains of the smaller ones indicates that more dense connectivity or a learned connectivity pattern is not necessary. They would increase complexity and slow down the model.

---

> > > > ### Comment · Area_Chair_xQFn · 2024-08-10
> > > > **Thank you for the valuable inputs**
> > > >
> > > > I understand reviewer Ljsg's point that results useful for small models might not directly apply to billion-scale or trillion-scale models. However, we should also recognize research that advances the foundational understanding of models and their scaling. Dense connectivity could potentially lead to higher compute per example, which in turn could improve data efficiency. I would appreciate seeing more theoretical and empirical evidence to support this claim.

---

> ### Author Response · Authors · 2024-08-12
>
> We thank reviewers 1YUF and EGgo for their responses and for their support. We also thank the AC for his comment.  We agree with these responses and hope they address reviewer Ljsg's concerns. We remain available to answer any further questions.
>
> Concerning the mention of 100B parameter models in the abstract, we will change it in our next revision.
>
> We also agree with reviewer 1YUF in that comparing scaling laws for DenseFormers and Transformers would be a convincing way to verify that our results generalize to larger model sizes. However, deriving those scaling laws requires running experiments at a scale that needs more computation than is available to us.

---

### Decision · Program_Chairs · 2024-09-25

**Decision:**

Accept (poster)

**Comment:**

Given the overall positive reviews, strong empirical results, and the authors' thorough responses to the reviewers' concerns, I recommend accepting this paper. DenseFormer presents a simple yet effective modification to the Transformer architecture, demonstrating its potential to improve the performance and efficiency of various models.